# Label Privacy in Split Learning for Large Models with Parameter-Efficient Training

## Abstract

As deep learning models become larger and more expensive, many practitioners turn to fine-tuning APIs. These web services allow fine-tuning a model between two parties: the client that provides the data, and the server that hosts the model. While convenient, these APIs raise a new concern: the data of the client is at risk of privacy breach during the training procedure. This challenge presents an important practical case of vertical federated learning, where the two parties perform parameter-efficient fine-tuning (PEFT) of a large model. In this study, we systematically search for a way to fine-tune models over an API *while keeping the labels private*. We analyze the privacy of LoRA, a popular approach for parameter-efficient fine-tuning when training over an API. Using this analysis, we propose $P^3EFT$, a multi-party split learning algorithm that takes advantage of existing PEFT properties to maintain privacy at a lower performance overhead. To validate our algorithm, we fine-tune DeBERTa-v2-XXLarge, Flan-T5 Large and LLaMA-2 7B using LoRA adapters on a range of NLP tasks. We find that $P^3EFT$ is competitive with existing privacy-preserving methods in multi-party and two-party setups while having higher accuracy.

## 1 Introduction

One of the main reasons behind deep learning success is its ability to transfer knowledge between tasks [34]. When training a model for any particular problem, it is common to reuse previously trained models from other, related problems. In the past, this was typically done by downloading pre-trained model weights from public hubs, then fine-tuning the said models on the downstream task. However, as models grow larger and more compute-intensive, fine-tuning them locally becomes an increasingly difficult task. Furthermore, many recent models are not released, but instead made available as proprietary services.

When a model cannot be fine-tuned locally, many practitioners opt instead for the so-called fine-tuning APIs [27, 16, 6, 26]. These APIs are web services that host one or several pre-trained models and allow clients to perform limited fine-tuning. More specifically, APIs usually allow their clients to run parameter-efficient fine-tuning (PEFT), such as LoRA [15] or Prefix-tuning [21]. These techniques allow adapting a model to a dataset while training a relatively small number of additional weights, which is particularly important for large language or image generation models that have billions of parameters.

Although the fine-tuning APIs can be convenient, they also introduce new risk in terms of data privacy. When a client uses such API to train on sensitive data, they need to ensure that their data will stay private [7]. This is particularly important when dealing with patient's medical records, personal user data or trade secrets [24, 19]. The two main threats to data privacy are that the API provider obtains the private data and that a third party intercepts data in transit. Therefore, data privacy is

not guaranteed even if the API provider is trusted. Several recent works propose LLM fine-tuning protocols that establish a certain level of privacy for multi-party fine-tuning [42, 7, 22]. Unfortunately, these algorithms work for a narrow class of fine-tuning algorithms or assume that a client can run LLM training locally using an obfuscated version of the model, provided by a remote server [42]. As a result, these algorithms are impractical for our use case of fine-tuning over an API. The few algorithms that are suitable for API fine-tuning guarantee the privacy of input tokens [22], meaning that the attacker can infer private training *labels*.

In this work, we seek to alleviate this problem by designing a two-party fine-tuning protocol that performs standard parameter-efficient fine-tuning with privacy guarantees. We formulate our protocol as a **special case of split learning** (or vertical federated learning), where one side (server) holds the pre-trained model and the other (client) has private training data. More specifically, we focus on **the privacy of client's training labels**. While input privacy is often crucial, there are scenarios where input data is publicly available, such as social media user pages. In these cases, labels could include ad clicks (known to the social network) or financial information (known to a bank that matches social profiles to its internal records). This example further justifies the use of LLMs, as social media pages often contain substantial amounts of text, and LLMs excel at processing long-context data.

Instead of developing a specific privacy-preserving architecture, we seek algorithms that can work with popular existing models and PEFT algorithms. Furthermore, our approach relies on the properties of parameter-efficient fine-tuning. Notably, since the adapters are compact, both parties can maintain multiple sets of adapters and swap between them with relative ease. This allows us to design a PEFT-specific algorithm that can solve its task more effectively than general split learning strategies [18].

We summarize our main contributions as follows:

- We analyze Low-Rank Adaptation, a common parameter-efficient fine-tuning algorithm, from the perspective of label privacy in the split learning setup. We observe that, despite fine-tuning less than $0.1\%$ of model parameters, PEFT algorithms leak client's training labels against simple attacks that work for modern pretrained transformers.

- Based on our analysis, we formulate a framework for privacy-preserving parameter-efficient fine-tuning ($P^3$EFT). This framework leverages the properties of PEFT to obfuscate the gradients and parameters communicated during fine-tuning with little impact on the fine-tuned model quality.

- To verify the practical viability of $P^3$EFT, we conduct experiments on popular real-world PEFT workloads[1]. Specifically, we fine-tune DeBERTa-v2-XXL [13], Flan-T5-Large [4] and LLaMA-2 7B [35] on a set of standard language understanding problems. We find that, compared to prior split learning algorithms, $P^3$EFT can maintain label privacy throughout training with a significantly smaller accuracy drop.

## 2 Background

### 2.1 Federated learning and split learning

Privacy preservation in machine learning has been a subject of active study within several frameworks. An important branch of privacy-preserving learning methods is federated learning, or FL [24], which can be broadly described as an approach allowing several parties to train a model jointly without sharing their private data. In particular, vertical federated learning [12, 43] targets the scenario where different features (including the label) of each training instance are kept by different parties.

One of the most popular approaches to vertical FL for neural networks is split learning [10, 37], where each party stores its part of the overall model. To train the model in such an approach, it is only necessary to transfer the intermediate activations and the gradients between layers, while the data itself is stored at the premises of the participant hosting each layer. In this work, we focus on the two-party formulation of split learning, where one side stores the features for each example and another one stores the labels.

---

[1]The code is available at `github.com/anonymousauthor56/P3EFT`

Recent works have investigated the setting of two-party split learning from the label leakage perspective [38, 28]: because the label party needs to pass the gradients of the loss function to the non-label party, it is possible for the latter party to deduce the labels by inspecting the gradients or activations or by hijacking the training procedure. Li et al. [18] provide a set of attack methods that allow recovering private labels and propose a defense mechanism that injects noise into the gradients; however, they test the approach on pretraining smaller models, and we study finetuning large models on private downstream data.

## 2.2 Parameter-efficient finetuning

The majority of large neural networks today are not trained with a specific task in mind: instead, they are pretrained on a general objective and then adapted for the downstream problem. Importantly, the growth in the size of foundation models has led to the increased popularity of parameter-efficient finetuning (PEFT) methods that adapt the model to a given task by training a small number of task-specific parameters. There are several prominent approaches to parameter-efficient finetuning, ranging from trainable prompts [21, 11], to residual adapters [14, 29]. We focus on Low-Rank Adaptation (or LoRA, 15), one of the most popular PEFT methods that adds extra parameters to each weight matrix in the form of a low-rank factorization (see Appendix C for a more detailed description). Such formulation allows LoRA adapters to be merged into the original weights after finetuning; this ability, combined with the simplicity of the method, has made LoRA a broadly popular approach in multiple domains. Still, the approach we propose can be applied to any PEFT method.

Several recent lines of work explore the problem of fine-tuning LLMs with privacy guarantees [44, 31]. Zhao et al. [46] analyze the viability of prompt tuning for federated learning, and Zhang et al. [45], Liu et al. [23] study PEFT algorithms in the setting of *horizontal* federated learning, that is, where multiple users train a shared model on their local private data. Another, more relevant research direction considers private fine-tuning in a *vertical* federated learning scenario, where participants hold different model layers [22, 40]. Most of these studies leverage the idea of differential privacy to prove an upper bound on how much information is leaked [8]. Unfortunately, these upper bounds are typically loose and do not match practical observations for real models. Furthermore, the majority of these studies only guarantees privacy of specific parts of the training procedure: for instance, Li et al. [22] only protects the input features, and not labels or model parameters. Finally, Xiao et al. [42] presents an alternative algorithm that protects client data by running the entire fine-tuning on client side by emulating the server-side model layers. While this approach is more holistic, it assumes that clients can run fine-tuning locally, which makes it impractical for many real-world users of LLM fine-tuning APIs. The primary distinction between our work and these studies is that we investigate parameter-efficient adaptation in the setting of split learning: we aim to finetune a model without disclosing the labels of examples to the model provider.

## 3 Privacy-preserving parameter-efficient fine-tuning

In this section, we analyze the privacy of parameter-efficient fine-tuning and propose a protocol for two-party parameter-efficient fine-tuning with the desired privacy guarantees. We begin by analyzing the privacy of API fine-tuning with popular PEFT algorithms in Sections 3.1 and 3.2. Then, in Section 3.3, we formulate a protocol for privately computing gradients over fine-tuning APIs. Finally, we formulate the full P$^3$EFT protocol in Section 3.4.

### 3.1 Setup

To analyze the privacy of API fine-tuning, we first need to formulate a common framework for this type of APIs and develop private learning protocols. This step is important, because existing fine-tuning APIs greatly vary in what they offer to the client: from closed APIs that require users to submit their full training data [27] to more flexible APIs where clients can run individual training steps [20, 2, 30]. Similarly to most existing works on split learning, we focus on the latter type of APIs that allows clients to run individual forward and backward passes over a remote model. Thus, a client can use these APIs to obtain the training gradients for their PEFT adapters, then update adapters locally with any optimization method. In our work, we adopt this archetype of fine-tuning API as it offers sufficient flexibility to develop privacy-preserving algorithms.

We formulate fine-tuning over an API for two or more parties: a client, and one or several servers. The client owns a training dataset with inputs $X$ and labels $Y$. In turn, each server has the same pre-trained model $h(x_i, \theta) \in \mathcal{R}^d$. Note that the parameters $\theta$ denote not the pre-trained model

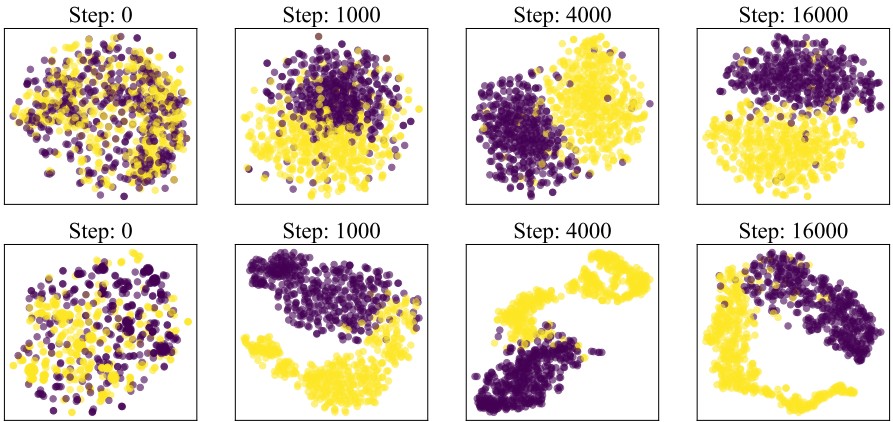

Figure 1: A visualization of top-2 principal components of gradients (top) and activations (bottom) from different fine-tuning steps (left to right). Color indicates the training labels (binary).

weights, but the trainable adapter weights for a certain PEFT algorithm. A model can encode an input $x_i \in X$ and produce a $d$-dimensional vector of activations that depend on the learned adapter weights $\theta$.

To allow fine-tuning, a server offers two API methods:

1. **forward**$(x, \theta) \to h(x, \theta)$ that computes model activations on input $x$ using adapter weights $\theta$;

2. **backprop**$(x, \theta, g_h) \to g_\theta$ that receives gradients of an arbitrary loss function w.r.t. model activations $g_h = \frac{\partial L(h(x,\theta))}{\partial h(x,\theta)}$ and returns the gradients w.r.t. adapter parameters, $g_\theta = \frac{\partial L(h(x,\theta))}{\partial \theta}$.

We further assume that both forward$(\cdot)$ and backprop$(\cdot)$ APIs are stateless and deterministic, i.e. calling the same API method multiple times (or on multiple servers) with the same inputs produces identical results. Thus, if the model uses dropout or any other form of non-determinism, we assume that clients provide the random seed as a part of $x$.

To fine-tune a model with this API, a client can initialize adapters locally, alongside with a small task-specific head[2], then train both adapters and the head. For each training batch $(x, y) \in D$, a client calls forward$(x, \theta)$ to compute feature representations, then predicts with local "head" and computes task-specific loss function $L$. After that, a client performs backward pass: first, it computes gradients w.r.t. local head inputs $g_h = \frac{\partial L}{\partial h}$, then passes those gradients to a remote server via backprop$(x, \theta, g_h)$ API call to compute gradients w.r.t. $\frac{\partial L}{\partial \theta}$. Finally, a client updates both $\theta$ and local "head" parameters using the optimizer of choice.

Before building more advanced algorithms, let us analyze the privacy of client's labels under standard fine-tuning. We consider an "honest, but curious" attacker model. This means that the server will faithfully run the forward and backprop computations as requested by the client without changing the results. Furthermore, we assume that servers are independent and do not communicate client's data between each other. However, a server can recover client's labels by performing arbitrary computations using any information it receives from the client. When training in this way, a client does not directly communicate training labels to the server. However, it communicates inputs, adapter parameters, and gradients. Furthermore, the server communicates input representations that can be intercepted by a third party.

## 3.2 Label Leakage of Standard Split Learning

In Figure 1, we train a DeBERTa-v2-XXL model on the SST-2 [32] sentiment classification dataset. The top row depicts the gradients $g_h$ communicated by the client when calling backprop$(\cdot)$ at different training stages. In the bottom row, we similarly track activations $h(x, \theta)$ that server may compute based on the specified $x, \theta$. We defer further additional figures and details to Section 4.1.

As we can see, both gradients and activations are arranged in such a way that simple k-means clustering would reveal which objects have the same label. The training activations (bottom row) do

---

[2]A linear layer that predicts class logits or regression target.

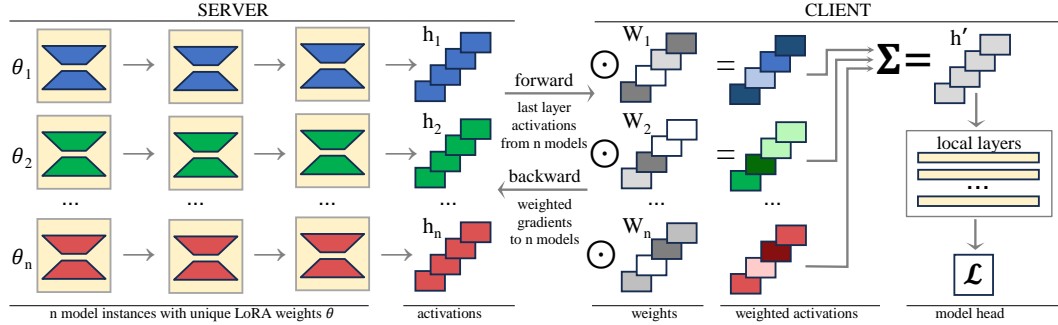

Figure 2: An intuitive illustration of the proposed fine-tuning protocol.

not reveal labels right away (at least not against this attack). However, they gradually "leak" private label information during training. Informally, it appears that the training gradients gradually pull apart the feature representations for each label, until eventually they turn into separate clusters. From an information-theoretic perspective, knowing just one vector of gradients *or* trained activations allows the attacker to learn all but one bit[3] of information about client's private labels.

To summarize, leaving any *one* data source unprotected (gradients, activations or parameters) would already compromise label privacy. However, we found that gradients and activations require different means of protection.

### 3.3 Privacy-preserving backpropagation

In this section, we formulate an algorithm for "anonymizing" the gradients communicated over a single training step with arbitrary PEFT type. Several prior works approach this by modifying the training objective or model architecture. However, when dealing with a real-world PEFT workload with optimized hyperparameters, changing the model or loss function often results in reduced model accuracy[4]. Thus, we seek an algorithm that preserves both model and training objective.

We design our algorithm based on an observation that **backpropagation is conditionally linear in output gradients**, even when the model itself is nonlinear. Formally, if we take a model $h(\cdot, \cdot)$, a fixed set of trainable parameters $\theta$ and input samples $x$, the backprop function[5] computes $\text{backprop}(x, \theta, \frac{\partial L}{\partial h(x,\theta)}) = \frac{\partial L}{\partial \theta}$. For convenience, we shorten it to $\text{backprop}(x, \theta, g_h) = g_\theta$, where $g_h = \frac{\partial L}{\partial h(x,\theta)}$ represents the gradients of some objective function with respect to model activations (outputs), and $g_\theta = \frac{\partial L}{\partial \theta}$ are gradients of the same objective function w.r.t. trainable parameters. In this notation, backprop is linear in terms of $g_h$ for any fixed $x, \theta$.

This becomes self-evident if we view backprop as multiplying $g_h$ by the Jacobian of model outputs w.r.t. trainable parameters, $\frac{\partial h(x,\theta)}{\partial \theta}$. If $x, \theta$ are constant, the Jacobian is also constant, and backprop is a linear operator:

$$\text{backprop}(x, \theta, \frac{\partial L}{\partial h(x,\theta)}) = \frac{\partial L}{\partial \theta} = \frac{\partial L}{\partial h(x,\theta)} \times \frac{\partial h(x,\theta)}{\partial \theta}. \tag{1}$$

This observation allows us to design a private backpropagation protocol. To illustrate this protocol, let us first consider a distributed API with two identical independent servers that offer backprop API. Then, for arbitrary vector $z$, we can rewrite $\text{backprop}(x, \theta, g_h)$ as $\text{backprop}(x, \theta, g_h + z) + \text{backprop}(x, \theta, g_h - z)$.

During API fine-tuning, we obtain $\text{backprop}(x, \theta, g_h + z)$ using an API call to server 1, whereas the second term $\text{backprop}(x, \theta, g_h - z)$ translates to an API call to server 2. Note that neither of two servers has access to the true gradient $g_h$: they only receive the sum $[g_h + z]$. If we sample a large noise vector $z$ ($\text{Var}(z) \gg \|g_h\|_2^2$), this sum also becomes dominated by noise. However, when both API calls finish, a client can sum the results to recover the true gradient of the loss with respect to parameters.

If both requests are processed by the same server, it can obviously recover $g_h$ by adding up gradients from both calls, which leads us to the final step. Instead of generating a single noise vector, a client

---

[3]The missing bit corresponds to attacker not knowing which cluster corresponds to label "1".

[4]We validate this empirically in 4.2.

[5]This is the same as the backprop API defined in Section 3.1.

needs to generate (privately) a set of $m > 1$ random vectors $\hat{g}_h^1, \ldots, \hat{g}_h^m$ and scalars $\alpha_1, \ldots, \alpha_m$ such that

$$g_h = \sum_{i=1}^{m} \alpha_i \cdot \hat{g}_h^i. \tag{2}$$

Then, for each $\hat{g}_h^i$, client computes $\text{backprop}(x, \theta, \hat{g}_h^i)$ as $m$ parallel API calls. Once this is done, client recovers

$$g_\theta = \sum_{i=1}^{m} \alpha_i \cdot \text{backprop}(x, \theta, \hat{g}_h^i). \tag{3}$$

Note that the client does not reveal $\alpha_1, \ldots, \alpha_m$ to anyone.

The resulting procedure is formulated in Algorithm 1. This algorithm is conceptually similar to the secure aggregation protocol for conventional (horizontal) federated learning [1]. This protocol allows clients to average their local vector with peers while keeping each individual vector provably private. Similarly to our scheme, clients perturb the vector in such a way that the average of perturbed vectors remains the same. Unlike Bonawitz et al. [1], our protocol privately backpropagates through a server-hosted model by leveraging the conditional linearity of the backpropagation operator.

---

**Algorithm 1** private_backprop — Privacy-Preserving Backpropagation (from the client's perspective)

1: **Input:** $x$ inputs, $\theta$ adapter weights, $g_h$ gradients w.r.t. activations, $m > 1$ - number of passes
2: $\hat{g}_h^1, \ldots, \hat{g}_h^m, \alpha_1, \ldots, \alpha_m = \text{obfuscate}(g_h, m)$ ▷ 2
3: **for** $j = 1, \ldots, m$ **do**
4: $\quad \hat{g}_\theta^j = \text{backprop}(x, \theta, \hat{g}_h^j)$ ▷ computed by server
5: **end for**
6: $g_\theta = \sum_{j=1}^{m} \alpha_j \cdot \hat{g}_\theta^j$
7: **Return:** $g_\theta$

---

The private backpropagation algorithm can allow client to safely compute gradients *once*, but, in practice, client usually needs to run many consecutive steps. This creates an additional vector of attack: if the same server receives two sets of parameters $\theta_t, \theta_{t+1}$, they could potentially recover $g_\theta$ by inverting the optimizer.

In the simplest case, if the server somehow knows that the client computes $\theta_{t+1} = \theta_t - \eta \cdot g_\theta$, then they can compute $g_\theta = (\theta_t - \theta_{t+1})/\eta$. While $g_\theta$ does not necessarily leak private labels, a server could, in some cases, use $g_\theta$ to recover $g_h$, either fully (e.g. if Jacobian is invertible), or partially.

The client has two ways to prevent this attack. The first one is to ensure that no single server runs backprop on two consecutive steps. This is easy to do in decentralized systems where there are many potential servers. However, even when there is a single server, they could be required to set up multiple trusted execution environments [25]. A more risky alternative is to ensure that the gradients cannot be reversed from consecutive parameters: randomize initial optimizer statistics or add noise to parameters. This solution is easier, but it can slow down training in some cases.

To summarize, we formulated a procedure that allows a client to compute gradients privately for any given model and PEFT type. Furthermore, since Equation 3 recovers true gradients, this obfuscation method does not affect the training dynamics. However, as we have shown in Section 3.1, gradients are not the only source of privacy leakage.

### 3.4 Full fine-tuning

The other major attack vector are training activations. As the model fits to training data, it's intermediate activations $h(x, \theta)$ allow attackers to recover labels, e.g. by clustering (see Figure 1). To combat this issue, we take advantage of the fact that PEFT has few trainable parameters. Instead of learning just one set of trainable parameters, a client creates $n$ independent adapter sets $\theta_1, ..., \theta_n$. Note that this does not require $n$ unique servers: a single server can run multiple sets of adapters. Furthermore, a client can alternate between using different servers for the same adapters. During forward pass, the outputs of different adapters are mixed together using randomized mixing weights $W \in \mathcal{R}^{n,d}$:

$$h'(x, \theta_1, \ldots, \theta_n) = \sum_{i=1}^{n} W_i \odot h(x, \theta_i) \tag{4}$$

Overall, we design this model in such a way the combined model $h'$ can predict the labels, but the adapters $h(x, \theta_i)$ do not allow predicting these labels without knowing the mixing weights W. The mixing weights are generated such that initial activations $h'(x, \dots)$ are equal to mean $h(x, \cdot)$ for all $x$. To achieve this, we generate W as follows: first, we generate $n \cdot (n-1)/2$ d-dimensional random vectors $\xi_{i,j} \in \mathcal{R}^d \forall i \in [1, n], j \in [i+1, n]$. Then, we add them up in the following way:

$$W = \begin{pmatrix} \frac{1}{n}e + \xi_{1,2} + \xi_{1,3} + \cdots + \xi_{1,n} \\ -\xi_{1,2} + \frac{1}{n}e + \xi_{2,3} + \cdots + \xi_{2,n} \\ \cdots \\ -\xi_{1,n} - \xi_{2,n} - \xi_{3,n} - \cdots + \frac{1}{n}e \end{pmatrix} \tag{5}$$

Here, $e$ stands for a vector of all ones. The purpose of these mixing weights is to ensure that the gradients w.r.t. individual $h(x, \theta_i)$ are obfuscated, but the averaged model behaves the same as regular PEFT adapter. To illustrate this, consider $n=2$ identical LoRA adapters $\theta_1, \theta_2$. During the first training step $h(x, \theta_1) = h(x, \theta_2)$. Therefore,

$$h'(x, \theta_1, \dots, \theta_n) = (1/2e + \xi_{1,2}) \odot h(x, \theta_1) + (1/2e - \xi_{1,2}) \odot h(x, \theta_2) = h(x, \theta_1) \tag{6}$$

However, the two adapters will learn different functions as they receive different gradients. From the first update on, $h'$ will be equal to an average of adapter predictions.

Finally, to ensure that individual adapters $h(x, \theta)$ do not accidentally "learn to leak" labels, we maintain this over the course of training with a privacy regularizer inspired by [9]. This ensures that it is impossible to predict labels from individual adapters $h(x, \theta_i)$. Intuitively, on each training step, client fits $n$ linear "heads" that learn to predict labels $y$ from $h(x, \theta_i)$, then performs an adversarial update of $\theta_i$ to prevent the "head" from predicting $y$. Formally, each of $n$ "heads" minimize the same objective function as the full model. For instance, if the full model solves multi-class classification, each head is trained to minimize cross-entropy:

$$\eta_i^* = \arg\min_{\eta_i} \sum_{x,y \in D} -y \cdot \log \frac{e^{\langle \eta_{ij}, h(x, \theta_i)\rangle}}{\sum_k e^{\langle \eta_{ik}, h(x, \theta_i)\rangle}}, \tag{7}$$

where y is one-hot encoding of the correct class.

The whole adversarial update takes place locally on client's side, using the same $h(x, \theta)$ it uses for the main training objective. The resulting procedure appears complicated but it typically takes negligible time compared to running the large pre-trainied model $h(x, \theta)$. Furthermore, since adversarial "heads" are linear, minimizing the objective above is done with standard logistic regression solver.

To summarize, our approach combines the two proposed ideas: we use the private backpropagation algorithm from Section 3.3 to protect the gradients, then trains a mixture of adapters in such a way that obfuscates learned activatons leaking labels. The resulting procedure is described in Algorithm 2. In the next section, we will evaluate the efficacy of P³EFT on popular NLP benchmarks.

# 4 Experiments

The main goal of our study is to find a practical method of private fine-tuning that would scale to large models. Because our approach leverages parameter-efficient fine-tuning techniques, we evaluate P³EFT with fine-tuning Transformer models on popular NLP benchmarks that these techniques were designed for.

To that end, we chose three pre-trained models: DeBERTa-XXLarge [13], Flan-T5-Large [4] and LLaMA-2 7B [35]. We train these models on several datasets from the GLUE benchmark [39]: SST-2 [32], MNLI [41] and QNLI.

## 4.1 Privacy of gradients and activations

For this experiment, we train DeBERTa-XXLarge on SST-2 dataset using LoRA adapters with hyperparameters from [15]. First, we train the model locally and track model activations $h$ and gradients w.r.t. those activations. We apply principal component analysis to them and plot the first

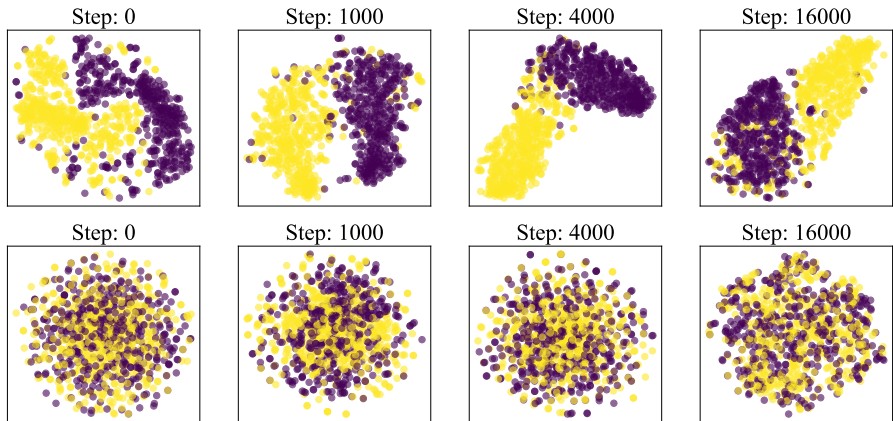

Figure 3: Gradients of cross-entropy w.r.t. LoRA parameters for DeBERTa-v2-XXLarge. The top row corresponds to normal backpropagation and the bottom row uses privacy-preserving backprop.

2 dimensions in Figure 1. Similarly, we visualize gradients of individual per-sample loss functions w.r.t. LoRA parameters $\theta$ in Figure 3 (top row). The results suggest that a hypothetical attacker could easily recover private labels by performing K-Means clustering over any data source: activations, gradients with respect to activations, or individual gradients with respect to parameters.

Next, we run the same experiment using privacy-preserving backpropagation as defined in Section 3.3. We use $n = 2$ with the noise variance set to 1000. As expected, we observed the same learning curve as with normal training. However, instead of sending gradients w.r.t. activations to the server, a client uses specially crafted random noise vectors that are not informative. In Figure 3 (bottom) we plot the same kind of individual gradients as in the top row, except that we visualize the gradients computed by the first of the two servers. Finally, we train XGBoost [3] with default hyperparameters to predict labels given the noisy gradients (pre-PCA): the resulting classifier is able to fit the training data perfectly, but has at most $50.4\%$ accuracy on a balanced test set.

## 4.2 Main fine-tuning experiments

Next, we evaluated the entire P3EFT algorithm. To control tasks and model type, we examined DeBERTa and Flan-T5 across all four datasets mentioned above, in addition to evaluating LLaMA on SST2 and QNLI datasets. For each setup, we compare against three baselines:

- **Without LoRAs.** In this baseline, the client gathers $h$ activations at the beginning (with no adapters), then proceeds to train local "head" layers using these activations. This method cannot leak information about training labels except for what is stored in X.

- **Regular fine-tuning (Regular FT)** refers to training a single LoRA adapter normally. This baseline represents an upper bound on model accuracy, but lacks privacy.

- **Distance Correlation (DC).** Our re-implementation of the distance correlation defense formulated in [33] for Transformer models.

For each algorithm, we evaluated a task-specific metric (accuracy or F1), as well as the privacy leakage value for the 3 following measures:

- **Spectral attack AUC** — a measure of vulnerability to an attack proposed in [33], measured as classifier ROC AUC: lower value corresponds to better privacy.

- **Norm attack AUC** — vulnerability to a variant of attack proposed in [18], measured as classifier ROC AUC (lower is better). Despite the initial proposal of this approach for attacking gradients, we observed that it is also well-suited for attacking activations.

- **K-means accuracy** — vulnerability to clusterization attack, measured in the percentage of correctly clustered activations, lower is better.

For all setups, we report the worst (least private) value among these metrics throughout the entire training period as a measure of privacy leakage, because it is the worst possible scenario that matters from the client's perspective. For DC and P³EFT, we specify the values for the best configuration in terms of the utility-privacy trade-off. See details in Appendix A. We also report adjusted standard deviations for the two privacy aware algorithms: P³EFT and DC. To do so, we run the full training procedure from scratch with 3 random seeds.

Table 1: Accuracy and privacy metrics. DeBERTa XXLarge.

| Dataset | | Without LoRAs | Regular FT | DC | P$^3$EFT |
|---|---|---|---|---|---|
| SST2 | *acc* | 82.9 | **96.9** | $96.6_{\pm0.4}$ | $96.5_{\pm0.2}$ |
|  | *leak* | **53.9** | 99.1 | $93.3_{\pm6.8}$ | $62.6_{\pm2.6}$ |
| QNLI | *acc* | 72.6 | **96.0** | $95.8_{\pm0.3}$ | $95.6_{\pm0.5}$ |
|  | *leak* | **51.5** | 99.1 | $85.0_{\pm11.6}$ | $74.6_{\pm11.1}$ |
| MNLI | *acc* | 49.2 | **91.9** | — | $86.9_{\pm0.5}$ |
|  | *leak* | **34.2** | 91.5 | — | $37.4_{\pm0.7}$ |

Table 2: Accuracy and privacy metrics. Flan-T5-Large.

| Dataset | | Without LoRAs | Regular FT | DC | P$^3$EFT |
|---|---|---|---|---|---|
| SST2 | *acc* | 92.8 | **96.1** | $95.0_{\pm0.1}$ | $\mathbf{96.1_{\pm0.1}}$ |
|  | *leak* | **55.8** | 98.3 | $68.1_{\pm5.0}$ | $74.1_{\pm3.0}$ |
| QNLI | *acc* | 83.2 | **95.3** | $95.2_{\pm0.1}$ | $94.7_{\pm0.0}$ |
|  | *leak* | **58.7** | 98.9 | $67.0_{\pm1.2}$ | $63.0_{\pm0.8}$ |
| MNLI | *acc* | 73.9 | **90.5** | $89.8_{\pm0.1}$ | $90.1_{\pm0.1}$ |
|  | *leak* | **34.6** | 85.9 | $45.6_{\pm0.8}$ | $40.0_{\pm1.1}$ |

The results for DeBERTa are presented in Table 1. To improve reproducibility, we reuse the hyperparameters from original paper, with the exception of the LoRA dropout value. We disable dropout because it interferes with the mixing weights (5). In preliminary experiments, we observed that with dropout enabled, both our algorithm and DC begin to perform significantly worse.

We use $n = 2$ adapter sets for P$^3$FT for all datasets and adhered to the same approach for the other models as well. Overall, P$^3$FT achieves nearly the same accuracy as traditional (non-private) fine-tuning, outperforming the DC-based algorithm in terms of accuracy given the same privacy level. On the MNLI dataset, we could not find the hyperparameters for DC that ensure stable training while maintaining privacy. Meanwhile, P$^3$EFT maintains consistent performance on this task with a slight drop in quality.

Table 2 a reports evaluation for the Flan-T5 base model[4]. For this model, we adapt the exact same hyperparameters as in the previous evaluation with DeBERTa-XXLarge. Compared to DeBERTa, these results are more closely matched. Both both our algorothm and DC consistently solve all three tasks, but P$^3$EFT slightly outperforms DC in terms of privacy.

Table 3: Accuracy and privacy metrics for LLaMA-2 7B.

| Dataset | | Without LoRAs | Regular FT | DC | P$^3$EFT |
|---|---|---|---|---|---|
| SST2 | *acc* | 94.6 | **97.4** | $97.1_{\pm0.1}$ | $95.8_{\pm0.1}$ |
|  | *leak* | **59.1** | 99.3 | $83.6_{\pm10.6}$ | $68.9_{\pm2.6}$ |
| QNLI | *acc* | 77.0 | 95.0 | $\mathbf{95.2_{\pm0.1}}$ | $94.7_{\pm0.2}$ |
|  | *leak* | **53.3** | 85.5 | $66.6_{\pm4.1}$ | $62.9_{\pm0.8}$ |

To evaluate how our algorithm scales to larger models, we also fine-tune Llama-2 7B [35] on SST2 [32] and QNLI [39] datasets. For these evaluations, we use LoRA hyperparameters that Hu et al. [15] used when fine-tuning GPT-3, with several changes inspired by Dettmers et al. [5]. Namely, we use the NF4 weight format, apply LoRA to both attention and MLP layers with rank 16. We fine-tune both tasks with maximum context length of 512 and weight decay 0.01. Table 3 summarizes our results: for QNLI, P$^3$EFT achieves somewhat better privacy-accuracy trade-off. On SST2, P$^3$EFT shows similarly favorable trade-offs while DC struggles to preserve privacy.

# 5   Conclusion and Discussion

In this work, we analyze privacy-preserving fine-tuning of large neural networks in the context of parameter-efficient fine-tuning and the two-party split learning setting. We show that while standard fine-tuning suffers from label leakage even in the parameter-efficient case, it is possible to leverage the efficiency of PEFT to alter the procedure without any significant performance drawbacks. We test the resulting method, named P$^3$EFT, on a range of pretrained language models and multiple datasets, showing that it is competitive with a strong baseline in terms of label privacy while having higher task performance.

In future work, it is natural to explore how this approach can be extended to establish holistic privacy in both labels and inputs. This problem can be approached from two directions: either adapt the ideas of P$^3$EFT for input privacy, or combine it with an existing work like [22]. Another important direction for future research is exploring the privacy of the long-term client-provider interaction. In a typical real-world use case of API fine-tuning, a client performs multiple training runs on overlapping data and hyperparameters. This could open additional attacks vectors that combine information from multiple training runs.

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

# A   Hyperparameters search

In P$^3$EFT and Distance Correlation methods resulting loss $L$ function can be viewed in the form

$$L = L_m + \alpha \cdot L_r,$$

where $L_m$ - main task loss, $L_r$ - regularizer and $\alpha$ is a coefficient that controls the tradeoff between these two losses. The selection of this coefficient affects the final performance of the model. Therefore, to find the best configurations for both methods, we iterated through this hyperparameter using a grid search.

We started with $\alpha = 1$ and then altered it with a multiplicative step of $10^{\frac{1}{2}}$. Values were discarded if the quality did not exceed that achieved by solely training the classifier without LoRA. This criterion was adopted because such outcomes would suggest the method's inability to outperform training scenarios in which the server does not engage with the labels whatsoever. Additionally, we excluded values that led to unstable training. By this, we mean instances where, although the model initially trained on the primary task, at some point, the regularizer began contributing significantly more, and the utility value dropped to the starting value. We observed this issue for the DC method with DeBERTa on the MNLI. From the remaining values, we aimed to choose the one that offered the lowest privacy leakage. The final hyperparameter values for P$^3$EFT can be found in the Table 4 and for DC in the Table 5.

Table 4: Regularization parameter $\alpha$ for the P$^3$EFT method. The values in the table represent powers of the $10^{\frac{1}{2}}$.

|                  | SST2 | QNLI | MNLI |
|------------------|------|------|------|
| DeBERTa XXLarge  | 1    | 1    | 1    |
| Flan-T5-Large    | -1   | 1    | 1    |
| LLaMA-2 7B       | 0    | 0    | —    |

Table 5: Regularization parameter $\alpha$ for the DC method. The values in the table represent powers of the $10^{\frac{1}{2}}$.

|                  | SST2 | QNLI | MNLI |
|------------------|------|------|------|
| DeBERTa XXLarge  | 0    | -1   | —    |
| Flan-T5-Large    | 2    | -1   | 0    |
| LLaMA-2 7B       | -1   | -1   | —    |

# B  Formal algorithm definition

Below, we define the full P$^3$EFT algorithm. In Algorithm 2, main_loss is the task-specific objective e.g. cross-entropy; reg_loss is the adversarial regularizer described in Section 3.4. We denote client-side model "head" as $f(h, \psi^t)$, where $\psi$ represent trainable head parameters. Finally, opt_step function performs a single gradient descent step with a task-specific optimizer, typically Adam [17].

---

**Algorithm 2** P$^3$EFT - full training algorithm

---

1: **Input:** dataset $D = \{X, Y\}$, $n > 1$ number of adapters, $\alpha \geq 0$ - regularizing weight, $m > 1$ number of obfuscated gradients
2: Initialize head $\psi^0$, mixing weights $W_i$ and adapters $\theta_i^0, i = \overline{1, n}$
3: **for** $t = 0, 1, \ldots, T - 1$ **do**
4:     Sample batch $\{x^t, y^t\}$
5:     **for** $i = 1, \ldots, n$ **do**
6:         $h_i^t = h(x^t, \theta_i^t)$                                                      ▷ by server
7:         $l_i = \text{reg\_loss}(h_i^t, y^t)$                                        ▷ by client
8:     **end for**
9:     $h' = \sum_{i=1}^n W_i \odot h_i^t$
10:     $l = \text{main\_loss}(f(h', \psi^t), y^t)$
11:     $L = l + \alpha \cdot \sum_{i=1}^n l_i$
12:     **for** $i = 1, \ldots, n$ **do**
13:         $g_h = \partial L / \partial h_i^t$                                         ▷ Client performs partial backprop
14:         $g_i^t = \text{private\_backprop}(x, \theta_i^t, g_h, m)$
15:         $\theta_i^{t+1} = \text{opt\_step}(\theta_i^t, g_i^t, t)$
16:     **end for**
17:     $\psi^{t+1} = \text{opt\_step}(\psi^t, \partial l / \partial \psi^t, t)$
18: **end for**
19: **Return:** $\psi^T, \theta_1^T, \ldots, \theta_M^T$

---

# C  Informal description of LoRA fine-tuning

For convenience, we provide a brief summary of fine-tuning with LoRA [15]. This PEFT method was originally designed for fine-tuning large pre-trained language models on downstream NLP tasks. These language models are typically based on the Transformer architecture [36], where most trainable parameters are allocated to linear layers in multi-head self-attention and feedforward blocks.

In the first stage of LoRA fine-tuning, user augments the model with adapters. To do so, a user goes over linear layers in transformer blocks and adds two trainable matrices, $A$ and $B$ that affect this layer's forward pass. Let $W_i \times x + b_i$ be the original layer with $n$ inputs and $m$ hidden units. Here, $W_i \in \mathcal{R}^{m \times n}$ is a pre-trained weight matrix, $b_i \in \mathcal{R}^m$ is a pre-trained intercept vector and $x \in \mathcal{R}^n$ represents a vector of inputs to this particular layer. During the forward pass, a layer with LoRA adapter computes $W_i \times x + b_i + B_i \times A_i \times x$, or equivalently, $(W_i + B \times A) \times x + b_i$. Here, $A_i$ and $B_i$ are two newly added matrices that constitute a LoRA adapter.

The adapter matrices $A \in \mathcal{R}^{r \times n}$ and $B \in \mathcal{R}^{m \times r}$ have a very small intermediate dimension $r$. For instance, when training GPT-3 with LoRA adapters, authors use $1 \leq r \leq 64$, whereas the main weight dimensions are $m = n = 12288$. The first matrix $A$ is initialized with small random normal values, and the second matrix $B$ is initialized at zeros. That way, initial $A$ and $B$ do not affect the model predictions.

Once all adapters are initilized, the user trains all $A_i$ and $B_i$ matrices of the model, while keeping the rest of the weights frozen. This way, only a small faction (less than 1%) of model weights are updated. Once the training is over, the learned adapters $A_i$ and $B_i$ can be merged into the main weights ($W_i := W_i + A_i \times B_i$) or used separately.

LoRA adapters are designed with two objectives in mind: i) to allow fine-tuning models in limited GPU memory and ii) to allow inferencing many fine-tuned models using one inference server. When fine-tuning, LoRA achieves small memory footprint due to the fact that user does not need to compute

gradients (or optimizer statistics) for billions of main model parameters. During inference, a server can keep a library of several adapters for different tasks and swap between them on demand.

## D    Informal description of LoRA fine-tuning

We used NVIDIA A100 GPUs for all the experiments. Experiments with DeBERTA [13] and Flan-T5 [4] on SST2 [32] were conducted on the single GPU, while experiments on MNLI [41] and QNLI require 4 A100. LLaMA-2 [35] expetiments were carried out on the node of 8 A100.

All the experiments last 12-24 hours. However, it is possible to speed up some of them using more GPUs, as well as conduct them on a smaller number of GPUs using technics to save GPU memory (see parameters in our code).

