# OpenReview forum: "Label Privacy in Split Learning for Large Models with Parameter-Efficient Training"
_NeurIPS.cc/2024/Conference — Submitted to NeurIPS 2024_

### Official Review · Reviewer_S8Y2 · 2024-06-21

**Soundness:** 3
**Presentation:** 3
**Contribution:** 3
**Rating:** 6
**Confidence:** 5

**Summary:**

The paper revisited the problem of label leakage in split learning in the context of fine-tuning large models with parameter-efficient training. Based on modern use cases, they proposed two privacy-preserving protections for gradients and activations during split learning. The proposed methods are evaluated on several large models including Llama2-7B, fine-tuned using LoRA and full model fine-tuning.

**Strengths:**

1. The paper moved forward a step on the urgent need to privacy-preserving split learning over large models and fine-tuning with LoRA.
2. The writing is generally good despite some minor issues. The flow of ideas is clear.
3. The proposed method is evaluated over different pre-trained large models, conforming to current real-world use cases of LLMs.

**Weaknesses:**

1. There exist several related works discussing the attacks and defense regarding the label leakage in split learning. The authors may need to compare the differences between the proposed methods and previous literature. The evaluation part lacks the comparison to existing privacy-preserving solutions over label leakage and some trivial solutions such as directly applying differential privacy, which is easy to implement.
2. In modern use cases of API fine-tuning, apart from the applications of classification, text generation with LLMs and image generation with multimodal models and diffusion are more common cases. And it is very critical to protect labels in these applications. For example, labels in text generation can contain answers to private questions in the private dataset. However, the leakage study and proposed privacy-preserving methods do not apply to these applications.
3. There are some minor writing issues that could be improved. For example, content introducing API fine-tuning and potential privacy concerns can be shortened in the introduction. The paragraph from line 53 to 58 can be reorganized so that it won't leave '[18]' for a whole line. Same thing for line 209. On line 28, write the full name of LoRA before using acronym. The authors are suggested to talk about split learning and no need to raise extra efforts for readers to understand what vertical federated learning is.

[1] Wan, Xinwei, Jiankai Sun, Shengjie Wang, Lei Chen, Zhenzhe Zheng, Fan Wu, and Guihai Chen. "PSLF: Defending Against Label Leakage in Split Learning." In Proceedings of the 32nd ACM International Conference on Information and Knowledge Management, pp. 2492-2501. 2023.
[2] Kariyappa, Sanjay, and Moinuddin K. Qureshi. "ExPLoit: Extracting private labels in split learning." In 2023 IEEE conference on secure and trustworthy machine learning (SaTML), pp. 165-175. IEEE, 2023.
[3] Erdoğan, Ege, Alptekin Küpçü, and A. Ercüment Çiçek. "Unsplit: Data-oblivious model inversion, model stealing, and label inference attacks against split learning." In Proceedings of the 21st Workshop on Privacy in the Electronic Society, pp. 115-124. 2022.
[4] Xu, Hengyuan, Liyao Xiang, Hangyu Ye, Dixi Yao, Pengzhi Chu, and Baochun Li. "Permutation Equivariance of Transformers and Its Applications." In Proceedings of the IEEE/CVF Conference on Computer Vision and Pattern Recognition, pp. 5987-5996. 2024.

**Questions:**

1. From my understanding, split learning usually involves multiple clients and a server. But according to line 137, it is a client and several servers. Can the authors explain about this?

**Limitations:**

Discussed in the last section.

---

> ### Author Rebuttal · Authors · 2024-08-07
>
> We are grateful for the reviewer's insightful feedback. We address their concerns and questions below.
>
> >The evaluation part lacks the comparison to existing privacy-preserving solutions over label leakage and some trivial solutions such as directly applying differential privacy, which is easy to implement.
>
> We are grateful to the reviewer for bringing to our attention the overlooked baseline from paper [1]. We conducted detailed experiments using this framework to train DeBERTa and Flan-T5 on the SST2 and QNLI datasets. In all setups, PSLF[1] performed notably worse than DC and P$^3$EFT. See details in the attached PDF (Table 1).
>
> >There exist several related works discussing the attacks and defense regarding the label leakage in split learning. The authors may need to compare the differences between the proposed methods and previous literature.
>
> We agree that the paper would benefit from a further discussion of prior art in label privacy attacks and defenses. Unfortunately, many of the prior works in label privacy are not feasible to use in our practical setup. We explain this in more detail below.
>
> From the perspective of label privacy the Split Learning framework has two potential vulnerabilities: activations and gradients. In attempts to compromise label privacy, the community has proposed numerous algorithms based on gradient exploitation, such as Cosine attack[2] and Norm attack[2], as well as more sophisticated methods like the attacks from UnSplit[3] and ExPloit[4] mentioned by the reviewer.
>
> However, our privacy-preserving backpropagation method allows us to avoid sending actual gradients to the server. Instead, the server receives vectors that appear indistinguishable from random noise, and **only the client knows how to reconstruct the true gradients** from them. This makes any gradient-based attacks (e.g. [2,3,4]) impossible when using privacy-preserving backpropagation. Therefore, in our work, we validate only those attacks that exclusively use activations. For the same reason we do not validate defense methods such as Marvell[2], as it was initially designed to protect against gradient-based attacks.
>
> We would also like to point out that our privacy-preserving backpropagation is independent of the second part of our method protecting activations and can be used in combination with many other methods proposed in the literature - e.g., together with DC, PSLF, or Differential Privacy methods.
>
> To better explain this in the paper, we will extend Section 2.1 to discuss these attacks, and then discuss how exactly private backpropagation protects against these attacks in Section 3.3.
>
> >In modern use cases of API fine-tuning, apart from the applications of classification, text generation with LLMs and image generation with multimodal models and diffusion are more common cases. And it is very critical to protect labels in these applications.
>
> We agree that the tasks of ensuring label privacy in the setups enumerated by the reviewer are of significant importance. However, each of these different setups has its own challenges that require different approaches, and it's not feasible to fit them all into one paper. For instance, text generation uses the same tokens as inputs and labels and the information can leak in both directions. Hence, adapting our method to this task would require combining it with some input privacy method.
>
> We've chosen to focus on classification as a specific, but important setup. This focus allows us to thoroughly examine key aspects of privacy-preserving techniques in split learning. We believe this provides valuable insights that can serve as a foundation for future research.
>
> >From my understanding, split learning usually involves multiple clients and a server. But according to line 137, it is a client and several servers. Can the authors explain about this?
>
> This is indeed different from a traditional use of split learning or vertical federated learning. However, these are the two closest research areas known to us, and as such, we treat our setting as an extension of the standard split learning setup. If the reviewer would like to suggest an alternative taxonomy, we would gladly consider it.
>
> > Xu et al., "Permutation Equivariance of Transformers and Its Applications."
>
> We thank the reviewer for bringing paper [5] to our attention. Although the authors mainly focus on data and model privacy in applications of their method rather than label privacy, this paper is relevant to our work due to the shared goal of achieving private client-server learning using large transformer-based models. We will include a discussion of this paper in the related work section.
>
> >There are some minor writing issues that could be improved.
>
> We are grateful to the reviewer for their invaluable contribution to improving the quality of the paper. We will certainly address all the shortcomings they have identified in the final revision.
>
> [1] Wan, Xinwei, Jiankai Sun, Shengjie Wang, Lei Chen, Zhenzhe Zheng, Fan Wu, and Guihai Chen. "PSLF: Defending Against Label Leakage in Split Learning." In Proceedings of the 32nd ACM International Conference on Information and Knowledge Management. 2023.
>
> [2] Oscar Li, Jiankai Sun, Xin Yang, Weihao Gao, Hongyi Zhang, Junyuan Xie, Virginia Smith, and Chong Wang. Label leakage and protection in two-party split learning. ICLR, 2022.
>
> [3] Erdoğan, Ege, Alptekin Küpçü, and A. Ercüment Çiçek. "Unsplit: Data-oblivious model inversion, model stealing, and label inference attacks against split learning." In Proceedings of the 21st Workshop on Privacy in the Electronic Society. 2022.
>
> [4] Kariyappa, Sanjay, and Moinuddin K. Qureshi. "ExPLoit: Extracting private labels in split learning." IEEE, 2023.
>
> [5] Xu, Hengyuan, Liyao Xiang, Hangyu Ye, Dixi Yao, Pengzhi Chu, and Baochun Li. "Permutation Equivariance of Transformers and Its Applications." In Proceedings of the IEEE/CVF Conference on Computer Vision and Pattern Recognition. 2024.

---

> > ### Comment · Reviewer_S8Y2 · 2024-08-08
> > **Response to the Rebuttal**
> >
> > Thank the authors for the rebuttal and complement experiments in tight time. My major concern about comparison with baselines  is complemented with experiments. Another question is can the authors explain why differential privacy is not compared or in other words, is there any particular reason that we do not have the necessity to compare with differential privacy?

---

> > > ### Author Response · Authors · 2024-08-11
> > >
> > > >can the authors explain why differential privacy is not compared or in other words, is there any particular reason that we do not have the necessity to compare with differential privacy?
> > >
> > > Our decision not to include differential privacy (DP) comparisons into the original paper was based on two factors:
> > >
> > > 1. As noted in [1] (Section 2), differential privacy and its variants are not directly applicable metrics in the context of split learning.
> > >
> > > 2. We found a lack of existing literature applying label differential privacy **specifically to split learning**, which presented challenges in identifying appropriate differential privacy baselines for comparison.
> > >
> > > However, during the review process, you brought to our attention the paper [2]. In this work, the authors developed a framework based on Randomized Response that satisfies Label Differential Privacy. We believe this method could indeed serve as a viable DP baseline that we had not previously identified.
> > >
> > > If the reviewer has any other suggestions for alternative DP baselines, we would be open to incorporating experiments with these in the final version of the paper.
> > >
> > > [1] Oscar Li, Jiankai Sun, Xin Yang, Weihao Gao, Hongyi Zhang, Junyuan Xie, Virginia Smith, and Chong Wang. Label leakage and protection in two-party split learning. ICLR 2022.
> > >
> > > [2] Wan, Xinwei, Jiankai Sun, Shengjie Wang, Lei Chen, Zhenzhe Zheng, Fan Wu, and Guihai Chen. "PSLF: Defending Against Label Leakage in Split Learning." In Proceedings of the 32nd ACM International Conference on Information and Knowledge Management, pp. 2492-2501. 2023.

---

> > > > ### Comment · Reviewer_S8Y2 · 2024-08-11
> > > >
> > > > Thank you for the explanation. I think it would be benefited to add this part related to DP into the future work, in the paper. As my primary concerns about comparison and DP are resolved, I raised the rate by 1, to 6.

---

### Official Review · Reviewer_T8zi · 2024-06-28

**Soundness:** 2
**Presentation:** 3
**Contribution:** 2
**Rating:** 5
**Confidence:** 3

**Summary:**

This study addresses the privacy concerns associated with the fine-tuning of Large Language Models (LLMs), focusing on SplitNN. It explores how gradients and activations can leak data, potentially allowing attackers to reconstruct original data sets. In experiments, the proposed method reduces label leakage while maintaining minimal utility loss.

**Strengths:**

S1. The manuscript highlights significant privacy issues in LLM fine-tuning, specifically the potential for data leakage through gradients and activations in SplitNN.

S2. Experimental results show that the proposed method significantly mitigates label leakage with minimal impact on utility.

**Weaknesses:**

W1. The claim that backpropagation is "conditionally linear" is not sufficiently rigorous. The manuscript suggests that $\text{backprop}(x, \theta, g_h+z)+\text{backprop}(x, \theta, g_h−z) = \text{backprop}(x, \theta, g_h)$ under the assumption that $\theta$ is constant. However, $\theta$ updates during each backpropagation, invalidating this assumption. Moreover, swapping the order of $\text{backprop}(x, \theta, g_h+z)$ and $\text{backprop}(x, \theta, g_h−z)$ could lead to different outcomes. Formal proof and a clearer statement of assumptions are needed to substantiate this claim.

W2. Section 3.4 describes a method to protect activations that resembles secure multi-party computation [1], lacking novelty. Its effectiveness is also questionable when only one adapter is present.

W3. The proposed protection mainly focuses on labels. In practice, data such as personal identifiers may pose a greater risk than labels. For example, knowing (a) Alice's salary (label) is included in the database is considered a more serious leakage than knowing (b) someone earns a salary of 3.2k. The manuscript should explore if the proposed method can also protect other sensitive features.

**References**

[1] Du, Wenliang, and Mikhail J. Atallah. "Secure multi-party computation problems and their applications: a review and open problems." Proceedings of the 2001 workshop on New security paradigms. 2001.

**Questions:**

See Weakness

**Limitations:**

There elaboration on limitations is insufficient.

---

> ### Author Rebuttal · Authors · 2024-08-07
>
> We thank the reviewer for their feedback and provide responses to their concerns hereafter.
>
> > The claim that backpropagation is "conditionally linear" is not sufficiently rigorous. The manuscript suggests that $\text{backprop}(x, \theta, g_h{+}z){ + }\text{backprop}(x, \theta, g_h{-}z) = \text{backprop}(x, \theta, g_h)$ under the assumption that $\theta$ is constant. However, $\theta$ updates during each backpropagation, invalidating this assumption. Moreover, swapping the order of $\text{backprop}(x, \theta, g_h{+}z)$ and $\text{backprop}(x, \theta, g_h {-} z)$ could lead to different outcomes. Formal proof and a clearer statement of assumptions are needed to substantiate this claim.
>
> We believe there has been a misunderstanding: the `backprop` API method **does not update $\theta$, it only computes the gradients** and returns them to the client (see line 146-148). Instead, the **client updates $\theta$,** which is possible because $\theta$ are parameter-efficient adapters.
>
> The detailed sequence of gradient computation and parameter updates is described in Algorithm 2 in Appendix B. Specifically, line 14 executes the `private_backprop` algorithm (during which the `backprop` API method is called multiple times). **After the gradients are computed** (and only then), the client updates the adapter set.
>
> If $\theta$ were updated during each `backprop` call , the equality would indeed be violated.
> For this reason, **we deliberately design `backprop` to be stateless**, i.e. it  merely computes gradients with respect to adapter parameters based on given gradients with respect to activations, without changing either the adapter weights or the weights of the original model.
>
> >Section 3.4 describes a method to protect activations that resembles secure multi-party computation [1], lacking novelty.
>
> Our approach is indeed related to the area of multi-party computation. However, we respectfully disagree that this affects our novelty. Multi-party computation is a broad research area that contains general cryptographic protocols (e.g. secret sharing). However, applying these protocols naively to the task of LLM fine-tuning would be inefficient, as it introduces an overhead of several orders of magnitude. (e.g. [8], page 7).
>
> The core novelty of our work is an efficient protocol specifically for the task of LLM fine-tuning. This protocol relies on the specifics of backpropagation (Section 3.3) to protect gradients and modifies the training procedure to protect the activations (Section 3.4). While we build on some prior ideas from general multi-party computation, we insist that our approach is novel.
>
> >The proposed protection mainly focuses on labels. In practice, data such as personal identifiers may pose a greater risk than labels. For example, knowing (a) Alice's salary (label) is included in the database is considered a more serious leakage than knowing (b) someone earns a salary of 3.2k. The manuscript should explore if the proposed method can also protect other sensitive features.
>
> We agree that the privacy of personal identifiers is highly important. Fortunately, this problem is broadly studied in prior work [1,2,3].
>
> In our work, we explore label privacy as one (but not the only) important component of private learning [4,5,6,7]. Since our approach is orthogonal to most feature privacy methods, it can be combined with these methods to protect both inputs and labels.
>
> > Its effectiveness is also questionable when only one adapter is present.
>
> Originally, we only considered training with multiple adapters for our experiments. To address your concern, we conducted additional experiments to better understand the effectiveness of our method with only $n=1$ adapter present.
>
> We report these experiments in Table 2 in the PDF and summarize them below . Surprisingly, we found that the $n=1$ setup demonstrates competitive performance on some (but not all) tasks. Still, training with a single adapter is less stable and, on average, inferior to the $n=2$ setup (see details in attached PDF). In future work, it is worth exploring single adapter setup further.  We will include this evaluation with additional discussion in the final version.
>
> We did our best to address your concerns with clarifications and additional experiments. We hope that this alleviates the weaknesses raised in the review and respectfully ask the you to re-evaluate your score. If you have any further suggestions (through review edits), we would be glad to apply them as well.
>
> [1] Yansong Li, Zhixing Tan, and Yang Liu. Privacy-preserving prompt tuning for large language model services. ArXiv, abs/2305.06212, 2023.
>
> [2] C. Song and A. Raghunathan, “Information leakage in embedding models,” ACM SIGSAC 2020.
>
> [3] Xudong Pan, Mi Zhang, Shouling Ji, and Min Yang. Privacy risks of general-purpose language models. IEEE Symposium on Security and Privacy.
>
> [4] Oscar Li, Jiankai Sun, Xin Yang, Weihao Gao, Hongyi Zhang, Junyuan Xie, Virginia Smith, and Chong Wang. Label leakage and protection in two-party split learning. ICLR 2022.
>
> [5] Jiankai Sun, Xin Yang, Yuanshun Yao, and Chong Wang. Label leakage and protection from forward embedding in vertical federated learning. arXiv:2203.01451, 2022.
>
> [6] Junlin Liu and Xinchen Lyu. Clustering label inference attack against practical split learning. arXiv:2203.05222, 2022.
>
> [7] Tianyuan Zou, Yang Liu, Yan Kang, Wenhan Liu, Yuanqin He, Zhihao Yi, Qiang Yang, and Ya-Qin Zhang. Defending batch-level label inference and replacement attacks in vertical federated learning. IEEE Transactions on Big Data, 2022.
>
> [8] Brian Knott, Shobha Venkataraman, Awni Hannun, Shubho Sengupta, Mark Ibrahim, and Laurens van der Maaten. Crypten: Secure multi-party computation meets machine learning. NeurIPS 2021.

---

> > ### Comment · Reviewer_T8zi · 2024-08-07
> > **Response to Rebuttal**
> >
> > I thank the authors for their thorough response. However, part of my concerns remain unaddressed.
> >
> > W1. I now understand that the statement is **correct** because parameters are not updated immediately after `backprop`. Despite this, I still recommend to
> > - highlight this staleness in Section 3.3
> > - modify Equation 1 as it does not reflect a linearity property. I guess you may want to put something like $\text{backprop}(a, b, c1) + \text{backprop}(a, b, c2) = \text{backprop}(a, b, c1+c2)$
> > - developing theoretical guarantee for this mechanism, as the accumulation of Gaussian noise can lead to some probability bound that guide the selection of #steps $m$. (This could be a significant contribution)
> >
> > W2. I agree that _applying secure multi-party computation (MPC) to split learning for label protection is novel_. However, applying an existing MPC approach is different from designing a new MPC approach. This paper appears to claim designing a new MPC approach because there is no citation or discussion of the used MPC mechanism. However, this proposed new mechanism must be compared with other MPC algorithms to support its novelty. The authors should clearly stated whether they are designing a new MPC method or using an existing one, followed by a comparison to the literature or clear citations.
> >
> > > The core novelty of our work is an efficient protocol specifically for the task of LLM fine-tuning.
> >
> > A followup concern for this response: Is this method specifically designed for LLM fine-tuning? In the original manuscript, it seems apply to all neural networks optimized by gradient descent.
> >
> > **Two-party case**:
> > > Originally, we only considered training with multiple adapters for our experiments
> >
> > Two-party case is not an exceptional case of multi-party split learning. Instead, the defense of the proposed approach is weakened as the number of parties decrease.The decreasing trend depends on the noise scale. A theoretically analysis is expected to demonstrate **what values $\mathcal{K}$ would likely invalidate the defense**.
> >
> > W3. I understand these are orthogonal topics, but my major concern is whether the focused scenario exists in practice. Can authors provide specific applications where labels urgently needs privacy protection?

---

> > > ### Author Response · Authors · 2024-08-11
> > >
> > > > Is this method specifically designed for LLM fine-tuning? In the original manuscript, it seems apply to all neural networks optimized by gradient descent.
> > >
> > > In the original manuscript, we note that our method leverages existing PEFT properties (e.g., **lines 11-12, 53-58, and 243**) and is applicable to any model trained with PEFT. Consequently, the scope is not limited to LLMs (for instance, PEFT is also applied to ViT[2] and diffusion models) or fine-tuning (e.g., see [3]). However, LLM fine-tuning remains the most prevalent application.
> > >
> > > Unfortunately, our method is not suitable for arbitrary neural networks. Specifically, the model must possess certain properties typical to PEFT — a large frozen model on the server side and a small number of trainable parameters. If you are interested in more detailed explanations, we would be happy to provide them.
> > >
> > > >**Two-party case**:
> > >
> > > We are unsure about your exact concern here, so we will try to respond to each point to the best of our understanding.
> > >
> > > >the defense of the proposed approach is weakened as the number of parties decrease
> > >
> > > We respectfully disagree with this statement. The experimental results with varying numbers of adapter sets ($n$) (see Table 2 in the attached PDF) **do not demonstrate this trend**. For instance, the results on SST2 with $n=4$ are inferior to those with $n=2$.
> > >
> > > >The decreasing trend depends on the noise scale.
> > >
> > > In Section 3.4, we employ noise solely in the coefficients $\xi$ in Eq. 5. Indeed, as the number of adapter sets ($n$) increases, the standard deviation of this noise increases as $\sqrt{n}$. However, the noise scale can also be directly amplified by increasing the standard deviation of the distribution from which $\xi$ is sampled. In our preliminary experiments, we attempted to vary the magnitude of the noise while keeping $n$ constant, but did not observe a clear correlation with the final results. If the reviewer is interested in this question, we can set up additional experiments to provide an answer.
> > >
> > > >A theoretically analysis is expected to demonstrate what values $\mathcal{K}$ would likely invalidate the defense.
> > >
> > > We did not use the notation $\mathcal{K}$ in our paper, but we assume this is a typo and the reviewer meant the number of adapter sets, $n$. We do not think that any particular value of $n$ will invalidate the defense. Even with $n=1$, the adversarial regularizer prevents the activations from pulling apart the feature representations for each label (unlike in Section 3.2). Our experiments conducted during the rebuttal phase with $n=1$ confirm this (for example, on SST2, the result with $n=1$ outperforms the DC baseline; see Table 2 for details).
> > >
> > > We would also like to emphasize that the rationale for using $n>1$ is described **in line 250**. To elaborate, we aim to create a situation where the activation distribution for each individual adapter is difficult to cluster or attack, yet the weighted sum of activations from Eq. 4 yields a distribution that is "simple" for subsequent classification. Achieving such a scenario for a pre-trained LLM with $n=1$ seems challenging, as we would simultaneously be training the same activation distribution to be both "simple for classification" and "difficult for clustering", whereas during conventional training, these properties typically correlate (see Figure 1).
> > >
> > > > Can authors provide specific applications where labels urgently needs privacy protection?
> > >
> > > One such example is provided in **Section 1, lines 48-52**. Additionally, there are several canonical examples where label privacy is crucial:
> > > 1. Advertising[4]. Consider an advertising platform A and an advertiser company B. Party A can record viewing history, while B possesses the visitor's conversion rate. B's labels remain private as purchases occur exclusively on their site or application.
> > > 2. Finance[5, 6]. An invoice agency A contributes invoice-related features, while a bank B provides credit data and SME labels. They collaboratively construct a risk model.
> > >
> > > [1] Bonawitz et al. Practical secure aggregation for privacy-preserving machine learning. ACM SIGSAC Conference on Computer and Communications Security 2017.
> > >
> > > [2] Dosovitskiy et al. An image is worth 16x16 words: Transformers for image recognition at scale. ICLR 2021.
> > >
> > > [3] Lialin et al. Relora: High-rank training through low-rank updates. ICLR 2023.
> > >
> > > [4] Li et al. Label leakage and protection in two-party split learning. ICLR 2022.
> > >
> > > [5] Cheng et al. Federated learning for privacy-preserving AI. Communications of the ACM, 63(12):33–36, 2020.
> > >
> > > [6] Hu et al. Is vertical logistic regression privacy-preserving? a comprehensive privacy analysis and beyond. arXiv preprint arXiv:2207.09087, 2022.

---

> > > > ### Comment · Reviewer_T8zi · 2024-08-11
> > > > **Response**
> > > >
> > > > I thank the authors for their thorough response. My concerns have generally been addressed. I have raise my rating to 5.
> > > >
> > > > For the theoretical guarantee in the response, I would recommend authors to provide more rigorous and formal probablistic bound like the definition of differential privacy.
> > > >
> > > > In addition, I suggest authors to check the privacy bound/proof of the existing MPC methods that are similar to their proposed one. This can improve the soundness of the proposed approach.

---

> ### Author Response · Authors · 2024-08-11
>
> >highlight this staleness in Section 3.3
>
> >modify Equation 1
>
> We thank the reviewer for their recommendations and will add these clarifications to the next revision of the paper.
>
> > developing theoretical guarantee
>
> Following the reviewer's recommendation, we provide theoretical analysis for privacy guarantees of the `private_backprop` algorithm. We use notations $B$ for `batch_size` and $d$ for `hidden_size`; $h_b, g_b, l_b$ correspond to activation, gradient and loss of the $b$-th batch element; $g_h$ represents a vector of concatenated gradients of all batch elements. We consider binary classification as a task with the minimum number of possible label combinations - $2^B$.
>
> We consider significantly stronger assumptions regarding the attacker's capabilities - namely, a white-box scenario. We assume the server knows the client-side model and, consequently, all possible $2^B$ vectors $g_h$ for different label sets. Thus, the server's task is to determine which of the $2^B$ label sets corresponds to the current batch based on the transmitted vectors.
>
> We investigate the minimum $m$ required to ensure that all $2^B$ sets remain equally probable from the attacking server's perspective in several possible setups:
>
> 1. Two non-interacting servers. In this scenario, it suffices to set $m=2$ and send one vector $\xi_i$ to each server $i$. From each server's perspective, all $2^B$ variants have equal probability because for any $\xi_i$ and a given $g_h$, there exists a vector $\eta$ such that $g_h$ belongs to the span of $\xi_i$ and $\eta$.
>
> 2. Single server. In this scenario, it is sufficient to set $m=B$.
>
>    To show this, we note that for the $b$-th element of the batch holds $\partial l_b/\partial h_b = \partial l_b/\partial p_b \times \partial p_b / \partial h_b$, where $p_b\in \mathbb{R}$ is the head's prediction for activations $h_b$ -  the probability of class 1. $\partial p_b/\partial h_b$ is a constant Jacobian of rank 1 and does not depend on the label value. Thus, both possible vectors $\partial l_b/\partial h_b$ lie in the Jacobian's image and belong to the same one-dimensional subspace.
>
>    Therefore, it is sufficient for the client to send a basis vector of the corresponding one-dimensional subspace $\partial p_b/\partial h_b$ for each batch element $b$ (and zero vectors for the remaining batch elements) one by one. Knowing $\alpha_b$, the client can reconstruct the corresponding contribution of $b$-th element to the weight gradient $g_{\theta}$. The server, however, cannot determine which label generated the given gradient for each example, as both lie on the same line.
>
> 3. Single server, $m < B$. In this setup, the client is not able to protect all gradients of the batch. Indeed, for $B=3$ and $m=2$, the set of $2^3$ possible gradient combinations cannot be embedded in any $2$-dimensional plane that the client can construct from $2$ vectors (the linear span of these $2^3$ gradients is $3$-dimensional). At most, the client can fully protect $m-1$ labels, while the server will know the remaining $b - m + 1$ labels up to a flip of all labels.
>
> We want to emphasize again that the above results obtained under a white-box assumption, which is significantly stronger than our practical setup. The general case is considerably more challenging for the attacking side, and developing a possible attack or determining the theoretical limits of the attacker's capabilities is a complex task. However, we believe that the theoretical analysis presented above may be a good first step in this direction.
>
> >This paper appears to claim designing a new MPC approach because there is no citation or discussion of the used MPC mechanism.
>
> Thank you for your comprehensive comment. We now have a better understanding of the reviewer's initial concern and will endeavor to provide a detailed response below.
>
> However, we believe there may have been a misunderstanding. **We do not state** that our method for protecting activations, described in Section 3.4 and referenced by the reviewer in W2 of the official review, pertains to MPC.
>
> Concurrently, we acknowledge that our method could be considered as belonging to MPC in a broader sense, as the `private_backprop` algorithm shares common ideas with this field. We **discuss this similarity in Section 3.3, L217-222, and cite prior work [1]** that leverages MPC.
>
> We have also used the phrase "multi-party" in the paper in reference to our method (L11, 15, 38), but **we have not explicitly mentioned MPC**. We employed these terms to highlight the potential benefits of a multiple-server setup when using `private_backprop` with fine-tuning API, which we discuss in detail in L223-235. If the reviewer disagrees with the use of the phrase "multi-party split learning," we are open to considering an alternative taxonomy.
>
> We hope our explanation has helped to resolve the misunderstanding. If the reviewer has any remaining questions regarding this concern, we would be glad to address them.

---

### Official Review · Reviewer_1WCr · 2024-07-03

**Soundness:** 3
**Presentation:** 3
**Contribution:** 2
**Rating:** 5
**Confidence:** 2

**Summary:**

This paper addresses privacy leakage during API-based Parameter Efficient Fine-Tuning (PEFT). Their designed P3EFT is a multi-party split learning algorithm that leverages PEFT adjustments to uphold privacy with minimal performance overhead. Their method proves competitive in both multi-party and two-party setups while achieving higher accuracy.

**Strengths:**

Researching API-based fine-tuning for large models is an intriguing topic, especially considering that many clients face challenges loading such large models due to size and computational constraints. In this scenario, privacy concerns regarding client data become paramount. This paper aims to mitigate potential privacy leakage by obfuscating gradients and parameters communicated during transmissions.

**Weaknesses:**

Their approach shows limited privacy improvement compared to the scenario  Without LoRAs, as indicated in Tables 1, 2, and 3, thereby restricting the overall benefits.

**Questions:**

1. What is the justification for using n=2 in line 332 for all the datasets?
2. How does the server benefit from this Split learning? How does the inference work after fine-tuning?
3. Address weaknesses.

**Limitations:**

The paper addresses limitations.

---

> ### Author Rebuttal · Authors · 2024-08-06
>
> We thank the reviewer for their feedback and answer their questions below.
>
> >Their approach shows limited privacy improvement compared to the scenario Without LoRAs, as indicated in Tables 1, 2, and 3, thereby restricting the overall benefits.
>
> The baseline 'Without LoRAs' represents the best-case scenario for privacy, since it does not transmit anything but input data. More advanced algorithms such as DC and P$^3$EFT (ours) cannot improve privacy compared to this scenario.
>
> However, training Without LoRAs means that the server-side model remains unchanged and cannot adapt to the task, leading to substantially worse accuracy. As a result, our method offers significantly higher accuracy (avg. 24.8% higher for DeBERTa)
>
> Ultimately, it comes down to the client's priorities - if privacy is paramount, they can opt for not training LoRAs, but if quality is more important and they're willing to accept a slight decrease in privacy, they can take advantage of the opportunity our method provides.
>
> >What is the justification for using n=2 in line 332 for all the datasets?
>
> Our preliminary experiments indicated that increasing $n$ barely improves the privacy of activations while introducing additional computational overhead. However, we agree that this matter should be explored further. To that end, we conduct additional experiments with various values of $n$ and include the results in the attached PDF (see Table 2).
> We thank the reviewer for this question and will add an extended ablation study on this hyperparameter in the final version of the paper.
>
> >How does the server benefit from this Split learning?
>
> We briefly touch upon this in Section 1 (line 25-31), but we agree that our paper would benefit from a more detailed discussion. To recall, our setup is based on the popular fine-tuning APIs, such as [1, 2, 3, 4, 5]. These APIs can be broadly split in two categories:
> * APIs for fine-tuning proprietary models (e.g. by OpenAI [1])
> * APIs for fine-tuning large open-source models (e.g. Hugging Face [2] or Replicate [5])
>
> In both cases, the API provider receives some form of compensation (i.e. money) for each use of their API. In turn, the clients benefit from the provider’s superior model (former type) or infrastructure (latter). By using privacy-preserving fine-tuning, the provider can reach more privacy-minded clients, which can increase their revenue, popularity, or similar.
>
> > How does the inference work after fine-tuning?
>
> The most obvious approach is to perform inference using the API's `forward` method, just as during training. Alternatively, several prior works [6, 7, 8] propose specialized algorithms for privacy-preserving inference that could be adapted to inference LLMs fine-tuned with our approach. We believe that this question is highly relevant to our work and plan on discussing it further in a separate subsection.
>
> [1] https://platform.openai.com/docs/guides/fine-tuning
>
> [2] https://huggingface.co/autotrain
>
> [3] https://octo.ai/docs/media-gen-solution/fine-tuning-stable-diffusion/fine-tuning-stable-diffusion
>
> [4] https://dreamboothapi.ai/
>
> [5] https://replicate.com/docs/guides/fine-tune-a-language-model
>
> [6] Edward Chou, Josh Beal, Daniel Levy, Serena Yeung, Albert Haque, and Li Fei-Fei. Faster cryptonets: leveraging sparsity for real-world encrypted inference. arXiv preprint arXiv:1811.09953, 2018.
>
> [7] Dacheng Li, Rulin Shao, Hongyi Wang, Han Guo, Eric P Xing, and Hao Zhang. Mpcformer: fast, performant and private transformer inference with mpc. arXiv preprint arXiv:2211.01452, 2022.
>
> [8] Jinglong Luo, Yehong Zhang, Jiaqi Zhang, Xin Mu, Hui Wang, Yue Yu, and Zenglin Xu. Secformer: Towards fast and accurate privacy-preserving inference for large language models. arXiv preprint arXiv:2401.00793, 2024.

---

> > ### Comment · Reviewer_1WCr · 2024-08-11
> >
> > Thank you for your thoughtful rebuttal. I appreciate the clarifications provided regarding the privacy trade-offs, particularly how the 'Without LoRAs' scenario represents the best-case privacy option and how your method balances this with significantly improved accuracy. The additional experiments on n=2 and your plan to include an ablation study in the final version address my concerns well. Your detailed explanation of the server’s benefits from split learning and the inference process post-fine-tuning also adds clarity. However, I will maintain my original score as the paper offers a valuable contribution, albeit with some trade-offs that merit further exploration.

---

### Official Review · Reviewer_Fveg · 2024-07-14

**Soundness:** 3
**Presentation:** 3
**Contribution:** 2
**Rating:** 5
**Confidence:** 3

**Summary:**

This paper proposes an algorithm to preserve the label privacy while achieve good accuracy in the split learning regime. The algorithm is used in parameter-efficient fine-tuning and empirically tested on some language models.

**Strengths:**

The paper clearly presents its motivation and contribution. The modification on the back-propagation is reasonable and empirically effective across three models and different attacks that are tested

**Weaknesses:**

The main concern is the scalability of this method. For one iteration, the number of backpropagation is m (at least 2), which is too slow even for PEFT. The computation cost of PEFT is 2/3 of full training so if m=2, the total cost of this method is 4/3 of full training.

Looking at the code, there are 7 hyperparameters introduced by this method, which may be hard to use in practice. I would suggest the authors fix some hyperparamters that the algorithm is robust to, to reduce the number of tunable hyperparameters.

Also the experiment results on SST2 show a severe leakage around 10% compared to without LoRA (even though this is relatively weaker than other methods).

**Questions:**

Have the authors considered other PEFT like linear probing and BiTFiT?

**Limitations:**

As in its current presentation, the method is limited to language models, split learning (two parties), and label privacy. The empirical evidence is limited to text classification (specifically, this method does not apply to natural language generation where LLAMA is originally trained for) and LoRA. Each limitation can be relaxed, e.g. extending to vision models, data reconstruction, additional PEFT, etc.

---

> ### Author Rebuttal · Authors · 2024-08-07
>
> We thank the reviewer for their feedback and address their concerns and questions below.
>
> >The main concern is the scalability of this method. For one iteration, the number of backpropagation is m (at least 2), which is too slow even for PEFT. The computation cost of PEFT is 2/3 of full training so if m=2, the total cost of this method is 4/3 of full training.
>
> Our approach does indeed introduce computational overhead, similarly to other privacy-preserving techniques. You are correct about per-iteration slowdown, but we would like to emphasize that model fine-tuning typically requires much fewer iterations than full training. For instance, LoRA adapters are known to fine-tune LLMs with 10-175B parameters in a matter of hours or days, using much less hardware than full training[1, 2]. Thus, we believe that our approach can be fast enough for some practical applications, even after accounting for the overhead.
>
> Other privacy-preserving fine-tuning methods also introduce considerable overhead. For instance, federated learning methods based on Homomorphic Encryption or Multi-Party Secure Computation achieve fully private training but incur overhead of several orders of magnitude (e.g. [5], page 7). Other existing methods, such as Distance Correlation [3] and methods based on differential privacy [4] offer less overhead at the cost of leaking some private information. Our approach belongs to the latter category, but offers a more favorable privacy-accuracy trade-off on the LLM fine-tuning tasks that we experiment with. That said, we agree that computational overhead is important and will discuss it more thoroughly in the final version of the paper.
>
> >Looking at the code, there are 7 hyperparameters introduced by this method, which may be hard to use in practice. I would suggest the authors fix some hyperparameters that the algorithm is robust to, to reduce the number of tunable hyperparameters.
>
> We believe there was a misunderstanding about our hyperparameters. We mention 7 hyperparameters only in the readme file from our supplementary code. However, **the actual number of hyperparameters is much smaller** – these 7 parameters include the options for running baseline algorithms.
>
> More specifically:
> * `coefs_method_name` is only needed for switching between baseline and our method.
> * `activation_lr_rw`, `shift_lr_rw`, `activation_dc_rw`, `shift_dc_rw` switch between our algorithm and the Distance Correlation (DC) baseline.
> * The `n_of_loras` **is** related to our algorithm, but we always use $n{=}2$ in all experiments (see line 332), so it is de facto constant.
>
> This leaves **only two hyperparameters:**  the regularization parameter $\alpha$ described in Appendix A (referenced line 325), and the magnitude of the noise used in Eq. (5). To further simplify hyperparameter tuning, we will add guidelines for setting these parameters in the final revision.
>
> >Also the experiment results on SST2 show a severe leakage around 10% compared to without LoRA (even though this is relatively weaker than other methods).
>
> This is a valid observation, as private fine-tuning still remains a challenging task. In our submission, we compare algorithms in terms of  trade-off between privacy and final quality.
>
> While fine-tuning without LoRAs does not leak privacy, it significantly compromises the accuracy of the main task. For example, not using LoRAs on DeBERTa loses 24.8% accuracy on average between 3 tasks.
>
> >Have the authors considered other PEFT like linear probing and BiTFiT?
>
> In principle, our approach can indeed be applied with other PEFT algorithms such as BiTFiT, since we only modify how the model is fine-tuned as a whole, without relying on any specific PEFT method. In our experiments, we chose LoRA as it is the most popular type of adapter used for NLP models. In turn, linear probing is identical to our baseline “Without LoRAs”. We thank the reviewer for this reference and will use this name in the updated paper.
>
> [1] Edward J Hu, yelong shen, Phillip Wallis, Zeyuan Allen-Zhu, Yuanzhi Li, Shean Wang, Lu Wang, and Weizhu Chen. LoRA: Low-rank adaptation of large language models. In International Conference on Learning Representations, 2022.
>
> [2] Tim Dettmers, Artidoro Pagnoni, Ari Holtzman, and Luke Zettlemoyer. Qlora: Efficient finetuning of quantized llms. Advances in Neural Information Processing Systems, 36, 2024.
>
> [3] Jiankai Sun, Xin Yang, Yuanshun Yao, and Chong Wang. Label leakage and protection from forward embedding in vertical federated learning. arXiv preprint arXiv:2203.01451, 2022.
>
> [4] Xinwei Wan, Jiankai Sun, Shengjie Wang, Lei Chen, Zhenzhe Zheng, Fan Wu, and Guihai Chen. Pslf: Defending against label leakage in split learning. In Proceedings of the 32nd ACM International Conference on Information and Knowledge Management, pages 2492–2501, 2023.
>
> [5] Brian Knott, Shobha Venkataraman, Awni Hannun, Shubho Sengupta, Mark Ibrahim, and Laurens van der Maaten. Crypten: Secure multi-party computation meets machine learning. NeurIPS 2021.

---

> > ### Comment · Reviewer_Fveg · 2024-08-12
> >
> > Thank you for your thoughtful rebuttal. It addresses most of my concerns. I have read it carefully and decided to maintain my score.

---

### Author Rebuttal · Authors · 2024-08-07

We would like to express our gratitude to all the reviewers for their detailed feedback.

We are pleased that the reviewers **1WCr** and **S8Y2** concur with our assessment regarding the critical importance of private API-based fine-tuning of large models in the contemporary landscape. We also appreciate **Fveg**'s and **S8Y2**'s acknowledgment of the paper's clear and coherent presentation. Additionally, we are glad that the reviewers **Fveg**, **T8zi** and **S8Y2** noted our empirical results.

The reviewers provided several valuable suggestions for improving the manuscript. Among others:
* Reviewer **S8Y2** highlighted PSLF[1] as a relevant baseline. In response, we conducted comprehensive experiments with this framework on the SST2 and QNLI datasets using DeBERTa and Flan-T5 models. Our findings, detailed in Table 1 of the attached PDF, indicate that PSLF[1] was consistently outperformed by Distance Correlation (DC) and P$^3$EFT across all four experimental setups.

* Reviewer **1WCr** inquired about the impact of the number of adapters on privacy and accuracy. To address this, we conducted experiments using the DeBERTa model on SST2 and QNLI datasets with $n=3$ and $n=4$ adapter sets. The results, presented in Table 2 of the PDF, generally demonstrate that the number of adapters has minimal influence on the resulting privacy and accuracy.

* Reviewer **T8zi** expressed interest in the efficacy of our setup when utilizing a single set of adapters. We examined this scenario using DeBERTa on SST2 and QNLI datasets. Despite slightly reduced stability concerning the $\alpha$ (reguralization weight hyperparameter), this setup proved highly competitive, which opens promising direction for further research. We have incorporated these results alongside previous experiments in Table 2.

We thank the reviewers for their insightful feedback and address the remaining questions and comments in our individual responses. We have also taken note of your observations and will address all the shortcomings in the final version of the paper.

[1] Wan, Xinwei, Jiankai Sun, Shengjie Wang, Lei Chen, Zhenzhe Zheng, Fan Wu, and Guihai Chen. "PSLF: Defending Against Label Leakage in Split Learning." In Proceedings of the 32nd ACM International Conference on Information and Knowledge Management, pp. 2492-2501. 2023.

---

### Decision · Program_Chairs · 2024-09-25

**Decision:**

Reject

**Comment:**

This paper proposes an MPC algorithm for protecting the privacy of labels when using parameter efficient fine tuning techniques in a split learning setting.

The reviewers agree that this work is highly important and that its motivation is clear. Further, reviewers have agreed the empirical results show the promise and efficacy of the approach. Yet, there are some key concerns that need to be addressed: i) in some cases, it appears that there is severe leakage compared with the baseline (“Without Loras”), ii) the per-step computational cost exceeds that of full training despite using PEFTs, iii) there are lacking formal guarantees, like DP, and iv) the setting of label privacy is perhaps a bit limited. I recommend the authors consider addressing these weaknesses.